# Spectroelectrochemical Properties and Catalytic Activity in Cyclohexane Oxidation of the Hybrid Zr/Hf-Phthalocyaninate-Capped Nickel(II) and Iron(II) tris-Pyridineoximates and Their Precursors

**DOI:** 10.3390/molecules26020336

**Published:** 2021-01-11

**Authors:** Yan Z. Voloshin, Semyon V. Dudkin, Svetlana A. Belova, Daniel Gherca, Dumitru Samohvalov, Corina-Mihaela Manta, Maria-Andreea Lungan, Samuel M. Meier-Menches, Peter Rapta, Denisa Darvasiová, Michal Malček, Armando J. L. Pombeiro, Luísa M. D. R. S. Martins, Vladimir B. Arion

**Affiliations:** 1Nesmeyanov Institute of Organoelement Compounds of the Russian Academy of Sciences, 119991 Moscow, Russia; voloshin@ineos.ac.ru (Y.Z.V.); sdudkin@ineos.ac.ru (S.V.D.); savkinasveta91@mail.ru (S.A.B.); 2Department of General and Inorganic Chemistry, Gubkin Russian State University of Oil and Gas (National Research University), 119991 Moscow, Russia; 3Sara Pharm Solutions S.R.L., 266-268 Calea Rahovei, 050912 Bucharest, Romania; daniel_gherca@yahoo.com (D.G.); dumitru.samohvalov@sara-pharm.com (D.S.); corina.manta@sara-pharm.com (C.-M.M.); maria-andreea.lungan@sara-pharm.com (M.-A.L.); 4Department of Analytical Chemistry, University of Vienna, Währinger Strasse 38, A-1090 Vienna, Austria; samuel.meier@univie.ac.at; 5Institute of Physical Chemistry and Chemical Physics, Faculty of Chemical and Food Technology, Slovak University of Technology in Bratislava, Radlinského 9, SK-812 37 Bratislava, Slovakia; denisa.darvasiova@stuba.sk (D.D.); michal.malcek@stuba.sk (M.M.); 6Centro de Química Estrutural, Instituto Superior Técnico, Universidade de Lisboa, Av. Rovisco Pais, 1049-001 Lisboa, Portugal; pombeiro@tecnico.ulisboa.pt; 7Institute of Inorganic Chemistry, University of Vienna, Währinger Strasse 42, A-1090 Vienna, Austria

**Keywords:** clathrochelates, phthalocyanines, zirconium(IV), hafnium(IV), nickel(II), iron(II), spectroelectrochemistry, homogeneous catalysis, oxidation reactions, DFT calculations

## Abstract

The in situ spectroelectrochemical cyclic voltammetric studies of the antimony-monocapped nickel(II) and iron(II) tris-pyridineoximates with a labile triethylantimony cross-linking group and Zr(IV)/Hf(IV) phthalocyaninate complexes were performed in order to understand the nature of the redox events in the molecules of heterodinuclear zirconium(IV) and hafnium(IV) phthalocyaninate-capped derivatives. Electronic structures of their 1e-oxidized and 1e-electron-reduced forms were experimentally studied by electron paramagnetic resonance (EPR) spectroscopy and UV−vis−near-IR spectroelectrochemical experiments and supported by density functional theory (DFT) calculations. The investigated hybrid molecular systems that combine a transition metal (pseudo)clathrochelate and a Zr/Hf-phthalocyaninate moiety exhibit quite rich redox activity both in the cathodic and in the anodic region. These binuclear compounds and their precursors were tested as potential catalysts in oxidation reactions of cyclohexane and the results are discussed.

## 1. Introduction

The covalent and coordination molecular assemblies with several electronically coupled metal centers are intensively studied [1,2,3,4,5,6,7,8,9,10,11,12,13,14,15,16,17,18,19,20,21,22,23,24,25,26] because of their prospective application in molecular electronics and light-harvesting (supra)molecular systems. Hybrid multicentered inorganic and organometallic transition metal arrays with relatively isolated π- and σ-electronic systems, which belong to different metal-centered frameworks, have received a relatively little attention up to date. These polytopic compounds have proven to be potentially useful building blocks for the design of redox- and photoredox-driven molecular electronic devices and artificial photosynthetic systems with long-lived charge separation states [27,28,29,30,31,32,33]. In particular, this class of hybrid transition metal complexes includes zirconium(IV) and hafnium(IV), as well as lutetium(III) phthalocyaninate-capped iron(II) cage compounds **1**–**4** (so-called “phthalocyaninoclathrochelates”), first prepared about 15 years ago [34,35] (Scheme 1). Their molecules contain a phthalocyanine-centered extensive π-system that is coupled with a quasi-aromatic metal-encapsulating macrobicyclic framework. These compounds have been characterized by single crystal X-ray diffraction, spectroscopic and electrochemical methods. Detailed spectroelectrochemical study of the oxidation of these hybrid complexes is also reported [36]. By comparison of their redox potentials with those of the parent zirconium(IV) and hafnium(IV) phthalocyaninates, the first reversible oxidation and reduction waves in the corresponding cyclic voltammograms (CVs) were attributed to the apical phthalocyanine moiety [37,38]. The second quasi-reversible and the third irreversible anodic waves were assigned to the oxidation of the clathrochelate framework based on similarity to the oxidation processes observed in the case of the parent iron(II) clathrochelates [39,40]. To corroborate the experimental electrochemical assignments and the theoretical density functional theory (DFT) calculations, the spectroelectrochemical experiments for the hybrid complexes **1**−**3** (Scheme 1) have been conducted [36]. Upon oxidation at their first oxidation potential, the intensities of the initial phthalocyaninate-localized *Q*- and *B*-bands decreased, while three new absorption bands appeared in the UV−vis spectra of these complexes. The oxidation was accompanied by several isosbestic points, thus suggesting formation of the corresponding phthalocyaninate-centered mono- and di-cation-radicals [41]. In contrast, upon electrolysis at the second oxidation potential, all the phthalocyaninate-centered UV–vis bands characteristic for their monocationic-radical forms lost their intensities without appearance of new absorption bands. The initial neutral iron(II) phthalocyaninoclathrochelates **1**−**3** could not be regenerated from the corresponding doubly oxidized macropolycyclic dications. This fact indicates [36] that the doubly oxidized dications are not stable and degrade in solution. Further evidence provided the CV data [35], which revealed quasi-reversibility of the second oxidation process only at high scan rates.

The redox properties of hybrid di- and trinuclear Fe,Zr- and Fe,Hf-porphyrinoclathrochelates **5**–**8** shown in Scheme 1, which are formed by capping with Lewis-acidic M^IV^ TPP group(s), where M^IV^ is Zr^4+^ or Hf^4+^ cation and TPP^2−^ is 5,10,15,20-tetraporphyrinate dianion, were also investigated [42] by various electrochemical and spectroelectrochemical techniques. The results of the CV and differential pulse voltammetry (DPV) experiments suggested that, in the case of the monoporhyrinoclathrochelates **5** and **6**, three redox processes could be clearly identified. Two reversible oxidation waves were assigned to the Fe^2+/3+^ and the TPP^1–/2–^ clathrochelate- and porphyrinate-localized redox processes, respectively, while the cathodic wave was attributed [42] to the TPP^2−/3−^ reduction. In the case of the iron(II) diporphyrinatoclathrochelates **7** and **8**, the Fe^2+/3+^ redox process is substantially shifted to lower potentials due to a more electron-donating effect of its porphyrinate ligand fragment, as compared with that of the capping boron atom in the monoporphyrinato-clathrochelate analogs. The porphyrinate-centered oxidation of these tritopic complexes was observed [42] as two closely separated but clearly recognized redox waves. Analogously, two closely spaced porphyrinate-centered reduction waves have also been detected [42]. In general, the UV–vis spectra of these di- and tritopic hybrid complexes suggest a lack of any electronic communication between the clathrochelate and porphyrinate frameworks. The corresponding (spectro)electrochemical data [42] also indicated very weak, if any, long-range electronic coupling between two macroheterocyclic π-systems in the molecules of trinuclear iron(II) diporphyrinoclathrochelates.

The phthalocyanine π-systems are known to possess photo- and electrochromic properties, as well as photocatalytic activity [43,44,45]. The phthalocyanine-based catalysts [46] can be easily obtained and possess very high chemical robustness even in harsh media. On the other hand, the designed *d*-metal clathrochelates were reported [47,48,49] to be useful for protein sensing or efficient electro- and (pre)catalysts for hydrogen and syngas (H_2_ + CO) production. A series of the polyamine cobalt(III) cage complexes with tethered mercaptoethylamide terminated oligopeptide apical substituents, allowing for immobilization on a surface of the working gold electrode, was described [50] to be potent heterogeneous electrocatalysts in both the hydrogen evaluation reaction (HER) and the oxygen reduction to hydrogen peroxide. Photodissociation of water was realized on rhodium-doped strontium titanate photoelectrode surface modified by a cobalt(II) clathrochelate when using methanol as a sacrificial agent [51]. Polyamine cobalt(III) clathrochelates were reported to be active and robust homogeneous catalysts for the oxidation of styrene by H_2_O_2_ both in acetonitrile solution and in ionic liquid medium [52]. The mononuclear cobalt(II) cage complex of a heteroditopic *N*_5_*O*_3_-macrobicyclic ligand was reported [53] to be an efficient catalyst for oxidation of a number of olefins and benzyl derivatives with molecular oxygen or 2-methylpropanal as the oxidants at room temperature and atmospheric pressure. Photoinduced catalytic oxygen production was realized in an aqueous acetonitrile in the presence of the mononuclear copper(II) cage complex with a fluorescent anthracene-containing polyamine macrobicyclic ligand as photocatalyst [54]. The unique water-stable iron(IV) clathrochelate [55] was discovered to catalyze the photochemical oxidation of water to dioxygen with high turnover frequency and with record turnover number, working at relatively low overpotential of this redox process [56].

Herein, we report on spectroelectrochemical studies of metal(IV) phthalocyaninate-capped iron(II) and nickel(II) tris-pyridineoximates along with their precursors shown in Scheme 2. In addition, the catalytic activity of **9**–**16** in homogeneous oxidation reactions of cyclohexane is also reported.

## 2. Results and Discussion

*Synthesis.* The heterodinuclear hybrid complexes **9**–**12** (Scheme 2) were prepared in good yields (65–79%) by transmetallation of the monotriethylantimony(V)-capped nickel(II) and iron(II) tris-pyridineoximates **13** and **14** as the reactive complex precursors with zirconium(IV) and hafnium(IV) phthalocyaninates [Zr(Cl_2_)Pc] (**15**) and [Hf(Cl_2_)Pc] (**16**) (see Scheme 2) as the Lewis acids in methanol–dichloromethane solutions/suspensions at room temperature [57]. The identity of the compounds **9**–**16** was verified by IR spectra (Appendix A) and ESI MS studies.

*ESI mass spectrometry.* Due to their cationic nature, the Ni- and Fe-containing compounds gave strong and clean single *m*/*z*-signals in the mass spectra. Their experimental masses and isotopic distributions corresponding to [M]^+^ matched the theoretical values on the full MS level (Table 1 and Figure 1). The Zr- and Hf-containing Pc precursors **15** and **16** released the chlorides and were detected as methoxido adducts [M − 2Cl + OCH_3_]^+^ (Table 1), which stemmed from methanol used as the solvent.

Each of the ionized compounds was then subjected to fragmentation experiments by collision-induced dissociation (CID). For this purpose, each precursor mass was selected and individually fragmented giving MS^2^ spectra. The Ni-Sb precursor **13** mass did not yield fragment ions on the MS^2^ level upon fragmentation. In contrast, the Fe-Sb precursor **14** showed ethyl group loss at the antimony, release of the chelating pyridineoximate ligand and oxidation (Appendix A).

Notably, the fragmentation spectra of **11** and **12** were similar, as were the respective fragment spectra of **9** and **10**. This indicates that the nickel(II) and iron(II) tris-pyridineoximates have a stronger impact on the fragmentation pathways of **9**–**12** than the respective group 4 transition metals zirconium(IV) and hafnium(IV) (Appendix A).

The MS^2^ spectra of the Fe-containing **11** and **12** (Appendix A) displayed characteristic fragments suggesting sequential release of the three pyridine ketimine moieties by cleaving the N–O bond and retaining O in the complex, i.e., Δm = 118. Furthermore, the fragment mass of 1082.2 of **12** might be indicative of FeO-release (Δm = 71). This was not observed for the iron analogue **11**.

Again, the nickel(II) complexes **9** and **10** yielded less fragments for interpretation. The fragmentation spectra of the nickel-containing **9** and **10** (Appendix A) showed a lower number of fragments, but the mass differences to the precursor ion suggested that the first pyridineoximate ligand was cleaved off intact, Δm = 136. Second, the Δm = 116 indicates the release of a pyridine ketimine ligand accompanied by redox changes on Ni, which was not observed for the Fe complexes. Release of the group 4 metals zirconium and hafnium was not observed. Then, MS^3^ experiments were performed for the bimetallic complexes by selecting a specific mass of each MS^2^ spectrum, which was again subjected to fragmentation. The fragmentation energy of the MS^3^ level was slightly higher than at the MS^2^ level. However, the MS^3^ level did not reveal significant new information about the investigated complexes (Appendix A).

*Cyclic voltammetry, spectroelectrochemistry and DFT calculations.* The cyclic voltammograms of complexes **9**–**12** in acetonitrile (ACN)/*n*-Bu_4_NPF_6_ (Appendix A) showed two fully reversible reduction waves and three oxidation events. Similar behavior was recently described for the same complexes in dichloromethane [57]. In the anodic part, the first two nearly reversible redox waves overlap, leading to the double peak shape. The third oxidation step is less reversible. Analogous redox response for all four investigated complexes indicates the ligand based redox locus. The reversibility and redox mechanism in the region of the first two reduction peaks for complex **9 (Ni**-**Zr)** was further investigated by the in situ spectroelectrochemical UV-vis-NIR cyclic voltammetric experiments in ACN/*n*-Bu_4_NPF_6_ under argon atmosphere in a thin layer honeycomb spectroelectrochemical cell (Figure 2).

Even by decreasing the scan rate to 10 mV s^−1^, the cathodic reductions remain reversible and upon scan reversal the products formed upon reduction are fully re-oxidized back to the initial state. The recovery of the initial optical bands upon the voltammetric reverse scan confirms the electrochemical and chemical reversibility of the redox process at the first and the second reduction peak, respectively. It is worth noting that in the region of the first reduction peak, two new optical bands at 585 and 982 nm, with vibronic structure, emerged (Figure 3a).

In addition, a decrease in the intensity of the initial optical bands at 335 and 682 nm, characteristic of the phthalocyaninate unit, indicates that the reduction takes place mainly on this part of the complex. This reduction in the region of the first electron transfer was accompanied by several isosbestic points what is characteristic for the formation of the corresponding phthalocyaninate-centered radicals [41,58,59,60]. This was confirmed by in situ EPR spectroelectrochemistry where a single line EPR spectrum with a *g*-value of 2.0049 and a line width Δ*H*_pp_ = 7.7 G appeared upon cathodic reduction at the first reduction peak (see black trace in Figure 4), confirming large spin delocalization on the phthalocyaninate unit with an extensive π-system. Localization of the unpaired electron on the phthalocyaninate moiety has been further confirmed by the DFT calculations, as can be seen in the corresponding spin density distribution maps (Figure 5). 

In the case of **11** and **12**, the spin density is localized exclusively on the phthalocyaninate moiety, while in the case of **9** and **10**, the spin density is found to be localized on the phthalocyaninate part as well as on the central atom (Ni) and in its vicinity (Figure 5).

At the second reduction peak, the bands of the 1e-reduced form at 585 and 982 nm decreased and a new band at 534 nm appeared. This can be attributed to the 2e-reduced form of phthalocyaninate moiety (Figure 3b). According to the DFT calculations, the triplet state is energetically preferred over the singlet state in these 2e-reduced forms of **9**–**12**, and the corresponding spin density distribution maps are displayed in Appendix A. A similar UV-vis-NIR spectroelectrochemical response was found for **Fe**−**Zr** analogue **11**, as shown in Appendix A, and also for **12 (Ni**−**Zr)** (Appendix A)

In the region of the first oxidation double-peak, new optical bands at 544 and 852 nm were observed, as shown for **9 (Ni**−**Zr)** in Figure 6, with a simultaneous decrease in the intensity of the initial optical bands at 335 and 682 nm, again indicating that the oxidation takes place mainly on the phthalocyanine part of the complex.

Although the anodic oxidation is less reversible, as can be seen in the in situ cyclic voltammogram, it was possible to measure the EPR spectrum of the corresponding 1e- oxidized **9 (Ni**−**Zr)** with a *g*-value of 2.0018, confirming even more delocalized spin density distribution in comparison to the 1e-reduced state (see red trace in Figure 4). Calculated spin density distribution maps of the oxidized forms of **9**–**12** (Appendix A) are very similar to the ones of their mono-reduced forms (Figure 5).

To confirm the suggested locus of the redox processes, the corresponding molecular precursors **13**–**16** (Scheme 2) were studied in detail by cyclic voltammetry (Appendix A) and spectroelectrochemistry. The electronic structure of 1e-oxidized and 1e-reduced states was studied by electron paramagnetic resonance (EPR), UV−vis−near-IR spectroelectrochemistry, and density functional theory calculations.

One fully irreversible cathodic peak was found for metal(II) tris-pyridineoximate precursors **13** and **14** (Scheme 2) in ACN/*n*-Bu_4_NPF_6_ solutions (Appendix A) at much more negative reduction potentials, as observed for the hybrid complexes (see Appendix A), providing evidence that the first two cathodic waves correspond to the reduction of Zr/Hf-phthalocyaninate units **15** and **16** (Scheme 2), which are reduced at much less negative reduction potentials (Appendix A). Note that in the corresponding hybrids, much more reversible redox behavior was found compared to that of the precursors. So, the incorporation of metal(II) tris-pyridineoximates exerts a stabilization effect on the charged Zr/Hf-phthalocyaninate frameworks. Nevertheless, EPR spectroelectrochemistry confirmed the formation of phthalocyaninate centered both 1e-reduced and 1e-oxidized radicals, as shown in Appendix A for precursor **15**. Interestingly, the corresponding 1e-reduced form of **15** exhibits the hyperfine splitting, while the 1e-oxidized form shows the single peak pattern, indicating larger spin delocalisation over the ligand, as confirmed by DFT calculations (see Appendix A, down).

The reversible anodic peak for Fe(II) tris-pyridineoximate **14** in ACN/*n*-Bu_4_NPF_6_ at +0.36 V vs. Fc^+^/Fc (Appendix A) corresponds to the Fe^2+^/Fe^3+^ redox couple. During in situ UV-vis-NIR spectroelectrochemistry at 10 mV s^−1^ at the honeycomb platinum working electrode, a less reversible behavior was found, but upon reverse scan, a recovery of the initial optical band at 540 nm, characteristic for the Fe^2+^ state, was observed, indicating rather high stability of the Fe^3+^ form (Figure 7). This band was assigned to the Fe(d)—Ligand(π*) MLCT in the iron(II)-containing tris-pyridineoximate fragment [57], which indicates a noninnocent character of the ligand. This corresponds well to the shape of HOMO orbitals of **13** and **14** (see Appendix A and Figure 8, respectively) and is in agreement with the noninnocent character of the tris-pyridineoximate moiety.

The Ni(II) tris-pyridineoximate precursor **13** in ACN/*n*-Bu_4_NPF_6_ showed an irreversible oxidation at *E*_pa_ = 0.53 V vs. Fc^+^/Fc that could be assigned to the Ni^2+^/Ni^3+^ redox couple. Analogous irreversible intense cathodic peaks at *E*_pc_ = −1.93 V vs. Fc^+^/Fc found for both M(II) tris-pyridineoximates **13** and **14** can be assigned to the 2e-reduction leading to the irreversible decomposition of the complexes, as evidenced by UV-vis-NIR spectroelectrochemistry for **14** (Appendix A).

*Catalytic studies.* The catalytic activity of dinuclear zirconium(IV) and hafnium(IV) phthalocyaninate-capped nickel(II) and iron(II) tris-pyridineoximate complexes **9**–**12** and their precursors **13**–**16** was tested for the peroxidative oxidation of cyclohexane as a model reaction. In fact, cyclohexane oxidation into cyclohexyl hydroperoxide and its deperoxidation into cyclohexanol and cyclohexanone are among the most important processes in the chemical industry, with the KA oil (cyclohexanone (K) + cyclohexanol (A)) mixture being the substrate for subsequent oxidation into adipic acid using nitric acid [61].

The catalytic experiments were initiated at room temperature and then by conventional thermal heating up to 60 °C, using an aqueous solution of tert-butyl hydroperoxide (THBP, 70%) as oxidizing agent (in view of its lower handling risk), and a low catalyst amount in ACN (see experimental).

The heterodinuclear hybrid complexes **9**–**12** exhibited some activity at room temperature (e.g., up to 9% KA oil yield for the Hf-Fe complex **12** after 6 h reaction), which was significantly enhanced by performing the oxidation reactions at 60 °C for the same time (6 h). Further increase in the temperature was avoided for safety reasons. The KA oil yields obtained under the optimized mild conditions are depicted in Figure 9.

The Hf-Fe complex **12** led to the highest KA oil yield (up to 36.9%) at 60 °C after 6 h reaction, suggesting a synergic effect on the catalytic activity of the heterodinuclear species. In fact, per se, the iron and hafnium complexes **14** and **16** yielded maxima of 20.6 and 15.9% of KA oil, respectively. 

A similar synergism, although not so much pronounced, is observed for the Zr-Fe complex **11**, which exhibits a higher activity than those of the precursor iron (**14**) and zirconium (**15**) compounds (e.g., KA oil yields of 26.3, 20.6 and 11.1%, respectively, after 6 h at 60 °C). However, eventual synergic effects are much less significant in the cases of the Hf-Ni (**10**) and Zr-Ni (**9**) dimetallic complexes. This suggests that the synergism is mainly based on the combination of the metals and not of the corresponding ligands.

The neutral compounds **15** and **16** (hence without the potentially dangerous perchlorate anion) were further tested for the MW-assisted oxidation of neat cyclohexane and, at the above optimized conditions (60 °C), afforded up to 26% yield of cyclohexanol and cyclohexanone mixture (for complex **16**) after 3 h under irradiation (Figure 10). Thus, the microwave irradiation appears to enhance the catalytic activity of both **15** and **16**, leading to significantly higher KA oil yields (26.0 vs. 11.6% and 19.1 vs. 7.0%, respectively, for **16** and **15** after 3 h reaction) than those obtained under thermal heating (compare Figure 9 and Figure 10). This is in accord with the previously observed behavior of other catalytic systems [62,63,64], where the use of MW irradiation provided a much more efficient synthetic method than conventional thermal heating. Due to the perchlorate counteranion of compounds **9**–**14**, their catalytic performance comparison under MW conditions was avoided. 

To date, the use of hafnium or zirconium as catalysts for the oxidation of cyclohexane is scarce. The first mention, in 1990, is a patented [65] process for the production of adipic acid by contacting cyclohexane with air at 80–160 °C and 2–100 bar in the presence of acetic acid, that used Zr and Hf co-catalysts (0.001–1000 atomic ratio Zr:Co, Hf:Co or (Zr+Hf):Co), making it possible to avoid the requirement for expensive halogen resistant construction materials. Two decades later, zirconium-porphyrinic iron-organic frameworks, with tetrakis(4-carboxyphenyl)porphyrin (TCPP), were applied by Zhou et al. [66], as catalysts for the cyclohexane oxidation to KA oil in the presence of TBHP (in decane) at 65 °C, but no catalytic activity was observed for the Zr–porphyrinic (without iron) compound. The reactivity was attributed to the high-density of accessible active porphyrinic iron(III) centres within the porous framework. Transition metal exchanged alpha-zirconium phosphates, alpha-ZrP center dot M (where, M = Mn(II), Cu(II) or Fe(III)) were also tested [67] for the oxidation of cyclohexane in liquid phase with TBHP. The order of reactivity of alpha-ZrP center dot M was: alpha-ZrP center dot Mn(II) > alpha-ZrP centre dot Cu(II) > alpha-ZrP center dot Fe(III). A maximum of 6% conversion and 100% selectivity for KA oil was observed with the alpha-ZrP center dot Mn(II)/TBHP system after 5 h of reaction, which could be recycled three times. Very recently [68], the magnetic ZrFe_2_O_4_@SiO_2_-TCPP nanocatalyst afforded a maximum of cyclohexanone and cyclohexanol products of 33.6 and 18.9%, respectively, with the advantage of being recovered by a magnetic field.

In the above cases, Zr is not the sole metal in the catalyst composition which also includes another transition metal with known catalytic activity in cyclohexane oxidation. Remarkably, in our study, besides the Zr and Hf heterobimetallic (**9**–**12**) complexes, we were also able to successfully use Zr and Hf monometallic neutral complexes (**15** and **16**) as efficient catalysts (yields up to 26% under MW irradiation, see Figure 10) for the selective oxidation of cyclohexane to the cyclohexanol and cyclohexanone mixture. Note that the current cyclohexane oxidation is a low-efficiency industrial process, with conversions lower than 10% to ensure a selectivity of 80% for the cyclohexanol/cyclohexanone mixture [69]. Thus, our catalytic outcomes represent a significant achievement in catalyst design for the industrially important cyclohexane oxidation.

The performed catalytic experiments support that cyclohexane oxidation catalyzed by our complexes proceeds through a mechanism of radical nature [70] where the catalyzed decomposition of t-BuOOH with formation of t-BuO and t-BuOO radicals is crucial for the H-abstraction from cyclohexane, the beginning of its conversion through the radical pathway. Thus, the availability of reducible complexes (proved by the spectroelectrochemical and DFT studies) would be a key requirement for the oxidation to occur.

## 3. Materials and Methods

*General procedures and measurements.* All reactions were performed under inert atmosphere using the standard Schlenk techniques. All reagents were obtained from the commercial sources and used without additional purification. Hybrid complexes **9**–**12** and their precursors **13**–**16** were prepared as described previously [57]. The IR spectra of **9**–**14** (see Appendix A) were in accord with previously reported data [57]. **Caution:** Complexes containing perchlorate should be handled with care.

*Electrochemical, spectroelectrochemical and related EPR experiments*. A Heka PG310USB (Lambrecht, Germany) potentiostat with a PotMaster 2.73 software package served as the potential control in cyclic voltammetric studies. Cyclic voltammetric experiments in acetonitrile (ACN) in the presence of *n*-Bu_4_NPF_6_ supporting electrolyte (puriss quality from Fluka) were performed under argon atmosphere using a three-electrode arrangement with platinum 1 mm disc working electrode (from Ionode, Tennyson, Australia), platinum wire as counter electrode, and silver wire as pseudo-reference electrode. Ferrocene (from Sigma-Aldrich, St. Louis, MO, USA) served as the internal potential standard. In situ ultraviolet-visible-near-infrared (UV-vis-NIR) spectroelectrochemical measurements were performed on a spectrometer Avantes (Model AvaSpec-2048 14-USB2, Apeldoorn, Netherland ) in the spectroelectrochemical cell kit (AKSTCKIT3) with the Pt-microstructured honeycomb working electrode, purchased from Pine Research Instrumentation (Durham, NC, USA). Halogen and deuterium lamps were used as light sources (Avantes, Model AvaLight-DH-S-BAL). The cell was positioned in the CUV‒UV cuvette holder (Ocean Insight, Ostfildern, Germany). Optical spectra were processed using the AvaSoft 7.7 software package. Amperostatic electrochemical in situ EPR experiments were performed in acetonitrile solutions containing 0.5 mM of a given complex and 0.2 M of *n*-Bu_4_NPF_6_. The freshly prepared solutions were carefully purged with argon and inserted in a Varian electrolytic cell equipped with a large platinum mesh. The EPR spectra were measured in situ using an X-band EPR spectrometer EMX (Bruker, Karlsruhe, Germany) with 100 kHz field modulation. 

*Electrospray Ionization Mass Spectrometry*. The compounds **9**–**12**, as well as their precursors **13**–**16**, were dissolved in methanol to a concentration of 1–5 μM and directly injected into the mass spectrometer. The compounds were analyzed intact and after fragmentation by collision-induced dissociation (CID) on MS^2^ and MS^3^ levels. The analysis was performed on a Bruker AmaZon Speed ETD using trapControl Version 7.1 and Data Analysis 4.0 SP5 (Bruker Daltonics GmbH, Bremen, Germany). The compounds were infused at a flow rate of 3 μL min^−1^. The data were acquired in the positive ion mode employing the following parameters: capillary voltage −4.5 kV, nebulizer 3 bar, dry gas 5 L min^−1^, dry temperature 180 °C. The mass range was 100–2200. Fragmentation experiments by collision-induced dissociation (CID) were performed using amplification energies of 1.5–2.2 on the MS^2^ level and MS^3^ experiments using amplification energies of 2.8–4.5. The low-level cut-off was 20% of the precursor mass by default.

*Computational details.* Geometry optimizations of the studied complexes (and their precursors) were performed at the B3LYP [71,72,73,74] /LanL2DZ [75,76] level of theory using the Gaussian09 program package [77]. The vibrational analysis was employed to confirm that the optimized geometries correspond to the energy minima (i.e., no imaginary vibration presented). Visualization of the optimized structures and the spin density distributions was performed using the Molekel software suite [78].

*Catalytic studies.* The catalytic tests were performed at room temperature or at 60 °C in a thermostated Pyrex round bottom flask and in open atmosphere, or under microwave (MW) irradiation. The MW experiments were undertaken in a focused Anton Paar Monowave 300 microwave incorporating a rotational system and an IR temperature detector (Anton Paar GmbH, Graz, Austria), using a 10 mL capacity reaction tube with a 13 mm internal diameter. Complexes **9** to **16** were used. (**Caution**: the combination of perchlorates with many oxidizable substances may be explosive; dry perchlorates at elevated temperatures may be explosive!).

The peroxidative oxidation reactions were carried out as follows: 1–10 μmol of the complexes **9**–**16** were added to ACN (3.00 mL) with vigorous stirring, whereafter 2.50 mmol of cyclohexane (270 μL) and 5.00 mmol (690 μL) of tert-butyl hydroperoxide (TBHP, 70% aqueous solution) were added and the reaction solution was stirred for 0.5–3 h at the desired temperature (from room temperature to 60 °C). Blank tests were performed in metal complex-free conditions.

The products analysis was performed as follows: 90 μL of cycloheptanone (internal standard), 10.00 mL of diethyl ether (to extract the substrate and the organic products from the reaction mixture) and an excess of triphenylphosphine were added. The obtained mixture was stirred for ca. 10 min and then a sample was taken from the organic phase and analysed by gas chromatography (GC).

GC measurements were carried out using a FISONS Instruments GC 8000 series gas chromatograph with a flame ionization detector and a capillary column (DB-WAX, column length: 30 m; internal diameter: 0.32 mm) and the Jasco-Borwin v.1.50 software (Jasco, Tokyo, Japan). The temperature of injection was 240 °C. The initial temperature was maintained at 100 °C for 1 min, then raised at 10 °C/min to 180 °C and held (at this temperature) for 1 min. Helium was used as the carrier gas. All products obtained from the catalytic oxidation reactions were identified by their retention times (confirmed with those of commercially available samples) and their quantification was attained by the internal standard method.

## 4. Conclusions

The ESI mass spectrometric study of **9**–**12** revealed pronounced similarity in fragmentation behavior of **9** and **10** as well as **11** and **12**. In particular, a sequential release of the three pyridine ketimine moieties was typical for the last two complexes.

The investigated hybrid molecular systems **9**–**12** exhibit quite rich redox activity both in the cathodic (two reversible reductions) and in the anodic region (two reversible and one quasi-reversible oxidations). EPR spectroelectrochemistry and DFT calculations indicate that the first reduction is due to the formation of phthalocyaninate-centered radical, while the second can be attributed to the 2e-reduced form of phthalocyaninate moiety. The dinuclear zirconium(IV) and hafnium(IV) phthalocyaninate-capped nickel(II) and iron(II) tris-pyridineoximate complexes **9**–**12** and their precursors **13**–**16** were found to catalyze the peroxidative oxidation of cyclohexane into cyclohexanol and cyclohexanone. The catalytic activity of **9**–**12** was low at room temperature, but increased significantly at 60 °C. The Hf-Fe complex **12** showed the highest KA oil yield (up to 36.9%) after 6 h reaction time (Figure 9). The yield was higher than that achieved in the presence of iron and hafnium precursors **14** and **16** (20.6 and 15.9% of KA oil, respectively), implying a synergistic effect of the two metals in **12** on its catalytic activity. Less pronounced synergism was observed for other heterodinuclear complexes (**9**–**11)**, but to a markedly lesser extent. Further temperature increase was avoided because of safety reasons. The synthesis of similar complexes to **9**–**12** with other than perchlorate counteranions and their testing at higher temperatures deserves attention. In this context, it should be stressed that the neutral compounds **15** and **16**, in which the counteranion is absent, have shown significantly higher KA oil yields in the MW-assisted oxidation of neat cyclohexane at 60 °C for 3 h (26.0 vs. 11.6 and 19.1 vs. 7.0%, respectively, for **16** and **15**) compared to those obtained by conventional thermal heating (Figure 10). The performed catalytic studies are in agreement with the mechanism of radical nature and availability of reducible complexes confirmed by spectroelectrochemical and theoretical DFT calculations.

## Data Availability

Data available in a publicly accessible repository.

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
