# Peer review of "Spectroelectrochemical Properties and Catalytic Activity in Cyclohexane Oxidation of the Hybrid Zr/Hf-Phthalocyaninate-Capped Nickel(II) and Iron(II) tris-Pyridineoximates and Their Precursors"

_molecules, 2021, doi:10.3390/molecules26020336_

Round 1

Reviewer 1 Report

The manuscript reports the study on spectroelectrochemical and catalytic properties of the hybrid Zr/Hf-phthalocyaninate-capped Ni(II) and Fe(II) tris-pyridineoximates and their precursors, the synthesis of which were reported in the previous article of the authors. The study is carried out in a comprehensive way and article is well written with appropriate referencing and detailed analysis of obtained data, so I recommend the publication after minor revision:

- In the section Materials and Methods, it is written that the studied compounds were synthesized according to the known procedures and their structures were confirmed by IR spectra. However the IR spectra were not shown, so provide them, please, in the supplementary materials.

- It should be noted that the authors refer to the previously reported studies on the oxidation of cyclohexane with Zr and Hf-containing catalysts. However, it is not clear what are the advantages of the described compounds in comparison with the known ones.

- The oxidation of cyclohexane is usually carried out under solvent-free conditions. Have the authors tested their materials under such conditions?

Author Response

- In the section Materials and Methods, it is written that the studied compounds were synthesized according to the known procedures and their structures were confirmed by IR spectra. However the IR spectra were not shown, so provide them, please, in the supplementary materials.

Response: The IR spectra of compounds 914 have been introduced now as Figures S1–S6  (Supplementary Material).

- It should be noted that the authors refer to the previously reported studies on the oxidation of cyclohexane with Zr and Hf-containing catalysts. However, it is not clear what are the advantages of the described compounds in comparison with the known ones.

Response: As mentioned in lines 399 to 404, we were also able to successfully use Zr and Hf monometallic neutral complexes (15 and 16) as efficient catalysts for the selective oxidation of cyclohexane to the cyclohexanol and cyclohexanone mixture, whereas the previous literature results where on heterometallic complexes, in which Zr or Hf is not the sole metal in the catalyst composition. The catalyst included another transition metal with known catalytic activity in cyclohexane oxidation.

- The oxidation of cyclohexane is usually carried out under solvent-free conditions. Have the authors tested their materials under such conditions?

Response: The aerobic oxidation of cyclohexane is usually carried out in added solvent-free conditions, i.e., neat cyclohexane. However, the peroxidative (either with aqueous hydrogen peroxide or tert-butyl hydroperoxide solutions) is commonly performed in acetonitrile or in solvent-free conditions when microwave-assisted. Acetonitrile is chosen due to its chemical inertness under oxidation conditions. These conditions (acetonitrile as solvent for the thermal heating and neat cyclohexane for MW-assisted) were also used in this work. In particular, for the cationic complexes bearing the perchlorate counteranion, the absence of solvent could increase the experimental risk.

Reviewer 2 Report

Review of paper “Spectroelectrochemical properties and catalytic activity in cyclohexane oxidation of the hybrid Zr/Hf-phthalocyaninate-capped nickel(II) and iron(II) tris-pyridineoximates and their precursors” prepared by Yan Z. Voloshin, Semyon V. Dudkin, Svetlana A. Belova, Daniel Gherca, Dumitru Samohvalov, Corina-Mihaela Manta, Maria-Andreea Lungan, Samuel M. Meier-Menches, Peter Rapta, Denisa Darvasiová , Michal Malček, Armando J. L. Pombeiro, Luísa M. D. R. S Martins, and Vladimir B. Arion.

The manuscript molecules-1055535 is focused on preparation of hybrid molecular systems that combine a transition metal (pseudo)clathrochelate and a Zr/Hf-phthalocyaninate moiety and their characterization by several techniques, such as cyclic voltammetric, UV-vis-NIR, EPR, ESI mass spectrometry, as well as modeling simulation using Gaussian and catalytic studies in oxidation reactions of cyclohexane. The paper reviewed contains interesting experimental results and their interpretation. However, some aspects should be discussed in more detail. In my opinion this article is worth to be published in Molecules after minor revision. I have some suggestions and questions, which author may consider prior to publication of this work:

  1. According to the Guide for Authors three to ten pertinent keywords should be added. Therefore, the authors must reduce the number of keywords in the manuscript.
  2. Authors wrote that “Complexes 9 to 16 were used. (caution: the combination of perchlorates with many oxidizable substances may be explosive; dry perchlorates at elevated temperatures may be explosive!”. If materials are risky, the pros and cons of using them must always be considered. The authors should indicate in the manuscript why they have decided on such a group of compounds.
  3. In section 3 – Results and discussion. Synthesis it is worth adding the yield in quantitative terms and not by indicating that it is high. Moreover, the main conclusions of the IR tests should be presented in the manuscript or in Supplementary Material.
  4. Explanations of the colors used in the Figures 3, 6, and 7 should be added to the text for easier interpretation of the results.
  5. The catalytic experiments should be discussed in more detail, for example yield of KA oil should be compared with typical catalyst using in oxidation reactions of cyclohexane. Such an approach would allow for a proper assessment of new hybrid materials in the context of their potential use in chemical processes.

Author Response

According to the Guide for Authors three to ten pertinent keywords should be added. Therefore, the authors must reduce the number of keywords in the manuscript.

Response: The number of keywords has been reduced to ten in accord with Editorial guidelines and recommendation of the reviewer.

Authors wrote that “Complexes to 16 were used (caution: the combination of perchlorates with many oxidizable substances may be explosive; dry perchlorates at elevated temperatures may be explosive!”. If materials are risky, the pros and cons of using them must always be considered. The authors should indicate in the manuscript why they have decided on such a group of compounds.

Response: As mentioned in lines 399 to 404, we were able to successfully use Zr and Hf monometallic neutral complexes (15 and 16) as efficient catalysts for the selective oxidation of cyclohexane to the cyclohexanol and cyclohexanone mixture. The previous literature data were on heterometallic complexes, in which Zr or Hf is not the sole metal in the catalyst composition. The catalyst included another transition metal with known catalytic activity in cyclohexane oxidation. The aerobic oxidation of cyclohexane is usually carried out in added solvent-free conditions, i.e., neat cyclohexane. However, the peroxidative (either with aqueous hydrogen peroxide or tert-butyl hydroperoxide solutions) is commonly performed in acetonitrile or in solvent-free conditions when microwave-assisted. Acetonitrile is chosen due to its chemical inertness under oxidation conditions. These conditions (acetonitrile as solvent for the thermal heating and neat cyclohexane for MW-assisted) were also used in this work. However, for the cationic complexes bearing the perchlorate counteranion, the absence of solvent could increase the experimental risk, therefore all experiments were performed in acetonitrile. Provided that the required safety measures were applied and taking into account that the available data on catalytic activity of this type of compounds in oxidation of cyclohexane are scarce, we performed the described experimental work.

In section 3 – Results and discussion. Synthesis it is worth adding the yield in quantitative terms and not by indicating that it is high. Moreover, the main conclusions of the IR tests should be presented in the manuscript or in Supplementary Material.

Response: The yields of the compounds in percent have been added now on p. 6 of the manuscript. The IR spectra of compounds 914 have been introduced now as Figures S1–S6 as supplementary material.

Explanations of the colors used in the Figures 3, 6, and 7 should be added to the text for easier interpretation of the results.

Response: Explanations required have been added to the legends of Figures 3 and 7.

The catalytic experiments should be discussed in more detail, for example yield of KA oil should be compared with typical catalyst using in oxidation reactions of cyclohexane. Such an approach would allow for a proper assessment of new hybrid materials in the context of their potential use in chemical processes.

Response: The suggested comparison was performed. Kindly see lines 403–407 in the text.